# Effects of Using Different Concentrate Supplementation Levels in Diets of Lambs Fed Tropical Aruana (*Megathyrsus maximus*) or Marandu (*Brachiaria brizantha*) Grass: Performance, Digestibility, and Costs of Production

Gustavo Daniel Vega-Britez [1,2], Marciana Retore [3], Allison Manoel de Sousa [4], Adrielly Lais Alves da Silva [2], Carolina Marques Costa [2], Carla Giselly de Souza [2], Marcio Rodrigues de Souza [5] and Fernando Miranda de Vargas Junior [2,*]

1 Facultad de Ciencias Agrarias, Filial Pedro Juan Caballero, Universidad Nacional de Asunción, Calle Lomas Valentinas y Rep. Cuba, Pedro Juan Caballero 130105, Paraguay; gustavo.vega@agr.una.py

2 Faculty of Agricultural Sciences, Federal University of Grande Dourados—UFGD, Itahum Rod, Km 12—Cidade Universitária, Dourados 79804-970, Brazil; drilais@hotmail.com (A.L.A.d.S.); carolinaufgd@hotmail.com (C.M.C.); carla.souza@esa.ipsantarem.pt (C.G.d.S.)

3 Embrapa Western Agriculture, Rodovia BR 163, Km 253 6 s/n Rural Zone, Dourados 79804-970, Brazil; marciana.retore@embrapa.br

4 Faculty of Administration, Accounting and Economics, Federal University of Grande Dourados—UFGD, Itahum Rod, Km 12—Cidade Universitária, Dourados 79804-970, Brazil; allison.msousa@gmail.com

5 Federal Institute of Mato Grosso do Sul—IFMS, Rua Filinto Müller, Dourados 79833-520, Brazil; marcio.souza@ifms.edu.br

* Correspondence: fernandojunior@ufgd.edu.br; Tel.: +55-67-99986-2018

**Abstract:** In Brazil, grazing is the main or only source of food for livestock. The appropriate combination of supplementation with concentrate in a lamb's diet on pasture is an alternative that can be explored to use natural resources to produce quality meat. The aim of the current study was to evaluate the effects of different supplementation levels (0%, 1.5%, and 3% of BW) on the intake, performance, and production costs of lambs grazing on Aruana (*Megathyrsus maximus*) and Marandu (*Brachiaria brizantha*) grasses. Thirty-six non-castrated male Suffolk lambs (22.54 ± 2.72 kg) were used. The lambs were evaluated for nutrient intake and digestibility, such as dry matter (DM), crude protein (CP), neutral detergent fiber (NDF), acid detergent fiber (ADF), and consumption in relation to body weight (% BW), in addition to the average daily gain (ADG), body condition score (BCS), and hot carcass weight (HCW), as well as the rising production cost of each system. The dry matter of the green leaf blades was influenced by the pasture type ($p < 0.05$), producing 1503.6 vs. 2977.4 kg/ha of Aruana and Marandu grasses, respectively. The concentrate supplementation level influenced the type of pasture ($p < 0.05$). A higher consumption of DM, CP, NDF, and organic matter was registered for the supplemented animals ($p < 0.05$) and on Aruana grass. The intake in relation to body weight was significantly influenced by the concentrate levels ($p < 0.05$). The empty body weight and HCW were significantly influenced by the supplementation levels ($p < 0.05$). The ADG and feed conversion (FC; kg DM/ADG) were influenced by the supplementation levels and type of pasture. The BCSs differed between the concentrate levels ($p < 0.05$). The supplementation improved dry matter digestibility. The ADG and FC were superior in the supplemented animals, with an advantage for those grazing on Aruana grass. The slaughter ADG was also higher in the supplemented animals. The lambs' pasture comprising Aruana grass with 1.5% BW of concentrate supplementation showed improved production and economic results.

**Keywords:** *Brachiaria brizantha*; feed efficiency; *Megathyrsus maximus*; sheep; tropical forage

## 1. Introduction

In Brazil, farmers utilize grazing as the main or only source of feed for their livestock. In the past, pastures with unproductive grasses characterized by low nutritional value were exclusively used, resulting in low rates of stocking and animal productivity. Increasingly, livestock producers are adopting grazing intensification, which has higher operating efficiency; using tropical grasses with greater productive potential has benefitted farmers, increasing their production capacity [1] and reducing production costs [2].

With the introduction of genera such as *Brachiaria* and *Panicum*, it was possible to select more productive materials adapted to local edaphoclimatic conditions [3].

Grazing animal efficiency depends on the intake and digestibility of the available pasture, which is directly related to the interaction between the canopy structure and the quantity and nutritional quality of the grazable green leaves [4].

Thus, higher amounts of green leaf blades are associated with higher animal performance due to changes in the quantity and quality of the diet available for selection, which is a performance that can be achieved more efficiently with the appropriate combination of concentrate supplements [5], with investment in supplementation being an important strategy to provide higher body weight gains [6].

Concentrate supplementation for sheep production is highly recommended. Crude protein (CP) has been identified as the most limiting nutrient in the tropics, especially during the dry season [7]. Protein supplementation is known to improve consumption and increase nitrogen supply to ruminal microorganisms.

This has a positive effect on the rumen microbial population and its efficiency, allowing an increase in the digesta degradation rate to be achieved. As the digesta degradation rate increases, intake consequently increases [8].

The appropriate level of concentrate supplementation in a lamb's diet on pasture is an alternative that can be explored to maximize the use of natural grazing resources to produce quality meat in addition to reducing production costs, allowing the farmer to have a greater resilience of responsiveness to the changing market requirements [9].

The economic return and good growth performance of lambs grazing on Aruana and Marandu grasses combined with concentrate supplementation are still lacking in Brazil. The objective of this study was to evaluate the intake, digestibility, performance, and economic analysis of lambs grazing on Aruana and Marandu grasses and receiving increasing levels of concentrate supplementation.

## 2. Material and Methods

### 2.1. Study Area Description

The experiment was carried out from December (Summer) to April (Autumn) at the Embrapa Agropecuaria Oeste Experimental Field in Ponta Porã City, Mato Grosso do Sul, Brazil (approved by the Ethics Committee on the Use of Animals of the Federal University of Grande Dourados—UFGD no. 027/2012). The experimental area is located between the geographical coordinates 22°32′56″ S and 55°38′56″ W, with an altitude of 642 m. According to the Köppen classification, the atmospheric conditions of the region fall under the Cwa climate type, defined as a temperate climate with a dry winter and rainy summer. The rainfall during the experimental period was 874 mm. The soil is classified as Red Latosol with a medium texture [10].

### 2.2. Animals and Supplementation

Thirty-six non-castrated, male, weaned Suffolk lambs were used, with an average age of 90 days and an initial average body weight (ABW) of 22.54 ± 2.72 kg. The lambs were housed in 12 paddocks measuring 32 × 32 m each. The treatments consisted of two types of pastures (Aruana and Marandu) and three supplementation levels (0%, 1.5%, and 3.0% of body weight (BW) based on the dry matter (DM) content).

The animals had access to water and mineral salt ad libitum throughout the experimental period. They were dewormed one (1) week before supplementation. The level of

worm infestation was monitored throughout the experimental period through parasitological fecal examination, measured as eggs/grams of feces (OPG), according to Gordon and Whitlock [11], at intervals of 21 days. As artificial shade, a 70% shade polyethylene screen measuring 3 m × 2 m was provided for each paddock. The average heights of the Aruana and Marandu grasses were 32.01 ± 7.45 cm and 51.16 ± 12.35 cm, respectively. The concentrate feed was composed of ground soybeans, corn, and oats, which was supplied daily at 8 a.m. Table 1 shows the pasture chemical compositions of the nutrients and concentrated feed that composed the lambs' diets.

**Table 1.** The proportion of ingredients in concentrate and chemical composition components of diets (%/kg of feed).

| Ingredients | Concentrate | Aruana | Marandu |
|---|---|---|---|
| (% in kg of feed) | | | |
| Oat grain | 45 | - | - |
| Soybean | 33 | - | - |
| Corn grain | 22 | - | - |
| **Nutrients** | **Chemical composition (mean and standard deviation)** | | |
| DM (%) | 87.1 ± 0.05 | 28.1 ± 0.84 | 31.5 ± 0.96 |
| CP (%) | 21.8 ± 0.08 | 16.6 ± 3.85 | 5.3 ± 1.36 |
| NDF (%) | 35.4 ± 0.10 | 62.9 ± 4.37 | 65.3 ± 6.23 |
| ADF (%) | 8.3 ± 0.04 | 29.8 ± 4.24 | 30.9 ± 5.21 |
| EE (%) | 8.8 ± 0.08 | 1.3 ± 0.59 | 1.2 ± 0.32 |
| MM (%) | 5.2 ± 0.01 | 7.8 ± 0.84 | 8.5 ± 1.03 |

DM: dry matter; CP: crude protein; NDF: neutral detergent fiber; ADF: acid detergent fiber; EE: ether extract; MM: mineral matter.

*2.3. Forage Evaluation and Measurement*

The grazing method used was continuous grazing with a fixed stocking rate (three experimental lambs per paddock). To estimate the total forage mass (kg/ha DM), the comparative visual estimation technique described by Haydock and Shaw [12] was used, with visual scores ranging from 10 to 30. These determinations were carried out every 28 days using a 0.25 m$^2$ area, totaling 24 sample points per paddock. During the total forage mass assessments, three forage samples by paddock were collected, in which grades 10 to 30 were first stipulated and then cut to generate a regression equation for the evaluation period.

The average value of the visual estimates of each experimental unit was considered as an independent variable in an equation of the type y = a + bx, where the visual estimates were related to the real value obtained by cutting and weighing. During the experiment, the grass height was measured from the ground to the curvature of the last leaf with the aid of a graduated scale of 150 cm.

The cut forage was homogenized and divided into two subsamples, one for DM determination and another for botanical separation (leaf, stem + sheath, and dead material). The samples were weighed and dried in an oven with forced air circulation at 60 °C for at least 72 h until a constant weight was reached. The percentage participation of the green leaf blade mass (GLBM), stem + sheath mass (SSM), dead material mass (DMM), green leaf mass/stem ratio mass (GLM/SRM), and the green leaf blade mass/dead material mass (GLBM/DMM) were determined using manual component separation.

The forage percentages of the GLBM, the stem mass, and the DMM, multiplied by the value of the GLBM in kg/ha, the stalk mass, and DMM, were obtained from the GLBM (kg/ha), stalk mass (SM, kg/ha), and DMM (kg/ha). The green leaf blade supply (GLBS, kg DM/100 kg BW) was calculated using a formula, GLBS = (GLBM n-1 + GBDAR) t-1 × 100, where GLBS = the green leaf blade supply; GLBM = the green leaf blade mass of each paddock/pasture (kg/ha DM); n = the number of days in the grazing cycle (days); GBDAR = the green blade daily accumulation rate (kg/ha DM); and t = the total number of animals in the period (kg/ha BW).

The GBDAR (kg/ha DM) was obtained using three grazing exclusion cages for each paddock/pasture type [13]. According to the following equation suggested by Campbell [14], GBDAR = $(C_i - OC_i - 1)/n$, wherein GBDAR = Green Blades Daily Accumulation Rate in period n (kg DM/ha/day); $C_i$ = DM inside the cages at instant i (kg DM/ha); $OC_i - 1$ = DM outside the cages at instant i − 1 (kg DM/ha); and n = number of days in the period, according to the equation suggested by Campbell [14]. These data were used to calculate the availability of the GBDAR (kg/ha DM), which was calculated using the forage mass (FM) arithmetic mean of the initial and final values for each experimental period of 28 days, divided by the number of days in the period, and then added to the corresponding GBDAR.

## 2.4. Feed Analysis

Forage samples were collected using a grazing simulation technique [15] to determine the chemical composition of the pasture. Concentrated food samples were defrosted at 28 °C, dried in a forced ventilation oven, and chemically analyzed. The DM was determined at 60 °C for 72 h, and the CP was calculated from the nitrogen content (N) (PB = N × 6.25) using the Dumas procedure (NSCH) according to AOAC [16]. The neutral detergent fiber (NDF) and acid detergent fiber (ADF) were analyzed using the method established by Van Soest et al. [17]. The ether extract (EE) was quantified using the acid hydrolysis method, as described by AOAC [16], and the ash content was measured gravimetrically via incineration in a muffle at 550 °C for 2 h after temperature stabilization.

## 2.5. Feed Consumption and Digestibility

The digestibility test was carried out in three periods, with three days of total feces collection each, where one lamb per paddock was equipped with a fecal collection bag. The bags were emptied twice a day. After weighing the total daily fecal production of each animal, the feces were mixed, and a 5% subsample was taken to become a single daily compound for each paddock. Composite samples were packed in sealed plastic bags and stored at −18 °C until laboratory analyses. The same procedure was performed with feed samples.

Feces and feed samples were incubated in vivo in non-woven bags (TNT, 100 g/m$^2$) for 288 h in the rumen of a fistulated bovine animal kept exclusively on pasture, according to the methodology described by Krizsan and Huhtanen [18].

The incubated sample amount was 0.5 g for feed and feces. The remaining material from the incubation was subjected to extraction with an acid detergent, and the residue was considered ADFi. The apparent digestibility (ADp) of DM, OM, CP, NDF, and ADF was determined as the proportion of DM or ingested nutrients not recovered in the feces using a formula, ADp = $1 - c_{total\ diet}/c_{ADFi}$, where $c_{total\ diet}$ = concentration (g/g) of ADFi in the DM from the total diet intake (forage + concentrate), and $c_{ADFi}$ = concentration (g/g) of ADFi in the fecal DM [19]. The intake was estimated using the formula proposed by Lippke [20]: Intake (g/day) = Fecal Production (g/day)/(1 − ADp). ADp is expressed as a percentage of the DM.

## 2.6. Experimental Procedures and Sampling

The lambs were weighed every 14 days at 8 a.m. using an electronic scale (0.1 kg accuracy) to adjust feed intake and calculate the average daily gain (ADG). At the time of weighing, the body condition score (BCS) was evaluated, which was always performed by the same evaluator for consistency and using the technique described by Russel et al. [21] and Kenyon et al. [22], with scores ranging from 1 (very thin animal) to 5 (very fat animal); as the lamb lots reached a BCS between 2.5 and 3.0, they were targeted for slaughter.

The ADG (kg) was determined as the difference between the final and initial body weights (FBW and IBW, respectively) divided over the feeding days. Feed conversion (FC) was calculated using the relationship between DM consumed per day and ADG and total weight gain (TWG, kg) was calculated by the difference between the FBW and IBW.

*2.7. Slaughter Procedure*

The lambs were slaughtered based on having a body condition score (BCS) of 2.5–3.0 (scale 1–5) [21,22] or at 6 months of age, following a finishing/physiological pattern of tissue deposition [23] regardless of treatment, resulting in slaughters at 73, 77, 91, 98, 105, and 126 experimental days, with an average of six animals per day. The animals were slaughtered 110 km from the experimental site, in an experimental slaughterhouse at the Federal University of Grande Dourados.

The lambs were slaughtered according to the protocol established by the Brazilian Regulation of Industrial and Sanitary Inspection of Animal Origin Products (RIISPOA) [24]. All animals were slaughtered under the same conditions and in accordance with ethics and welfare laws.

The animals were desensitized using electric discharge (electronarcosis). Then, the animals were bled using the carotid arteries and jugular vein sections, after which they were skinned, and the gastrointestinal tracts were removed and emptied to obtain the empty body weights (EBW = SW − gastrointestinal content). After evisceration, the carcasses were weighed to obtain the hot carcass weights (HCWs), whose market prices were used for economic analysis.

*2.8. Economic Analysis*

The direct costs of each production system were as follows: lease, lamb acquisition costs, mineral salt, dry mass of forage consumed, medicines, labor, and supplements. It is important to note that the supplement was not used in systems with 0% concentrate supplementation. In addition, other direct costs were attributed to the products, such as losses of DM and biological assets.

The indirect costs experienced an apportionment process, and the criterion for this procedure was the area used for each paddock, 1024 m$^2$, whereas the total area of the experiment was 12,288 m$^2$. The indirect production costs were electricity and fixed asset depreciation (fences, yard, furniture and utensils, and machinery and equipment). Expenses related to taxes were not considered, as the focus of this work was presenting the costs involved directly in the finishing of lambs.

The data from the identification, measurements, and appropriation were structured so that it was possible to analyze the proportion and degree of importance in each production cost composition. It was then possible to carry out the analyses and consequently evaluate which system had the most suitable performance. Twelve cost assessment items were calculated for the different treatments. First, we evaluated the direct cost, indirect and production (USD) costs, gross earnings (USD), gross profit (USD), gross margin (GM = %) per carcass produced (USD), cost per kg of carcass produced (USD), cost per concentrate (USD), average daily gross result (USD), and accounting breakeven point (kg and USD) [25].

*2.9. Experimental Design and Statistical Analysis*

A completely randomized design was used in a 3 × 2 factorial scheme with six replications per treatment (0%, 1.5% and 3% supplementation in relation to BW) and two types of pasture (Aruana and Marandu). Statistical analysis was performed with analysis of variance using the PROC GLM procedure (general linear model) from SAS [26]. The effect of the feeding system was considered as a fixed effect, and all variables were analyzed according to the following model:

$$Y_{ijk} = \mu + P_i + C_j + (P \times C)_{jj} + \varepsilon_{ijk} \tag{1}$$

where $Y_{ijk}$ = dependent variable; $\mu$ = overall mean; $P_i$ = the effect of pasture (*i*Aruana, Marandu); $C_j$ = the effect of concentrate supplementation (*j* = 0.0%, 1.5%, and 3.0%); (P × C) = the interaction effect of grass and supplementation; and $\varepsilon_{ijk}$ = the experimental error.

Whenever a significant difference at $p < 0.05$ was detected, a post hoc comparison test (Tukey's test) was performed. Pearson's correlation estimates between the variable

consumption and digestibility of nutrients and ADG were performed using the CORR procedure in the package [26]. The results were presented as mean ± standard deviation (SD).

## 3. Results

### 3.1. Quantitative Forage Estimate

There was no interaction effect ($p > 0.05$) between the treatments for the variable canopy structure of Aruana and Marandu grasses (Table 2). The pasture type had a significant effect ($p < 0.05$) on the following variables: the green leaf blade mass (GLBM) (kg DM/ha), green leaf blade supply (GLBS) (kg DM/100 kg BW), green leaf blade mass/stalk mass (GLBM/SM), and green leaf blade mass/dead material mass (GLBM/DMM). Conversely, the total forage mass (TFM) (kg DM/ha), SM (kg DM/ha), and DMM (kg DM/ha) were not associated with the provided pasture type, while the concentrate supplementation level had no effect ($p > 0.05$) on the variables presented.

**Table 2.** Total forage mass, green leaf blade mass, stalk mass, dead material mass, green leaf blade offer, green leaf blade ratio and stalk mass, and green leaf blade ratio and dead material mass from Aruana and Marandu grasses under different levels of concentrate supplementation.

| Pasture (P) | Concentrate Level (C) | | | Mean | *p*-Value | | |
|---|---|---|---|---|---|---|---|
| | 0% | 1.5% | 3% | | C | P | C × P |
| | \multicolumn Total forage mass (kg DM/ha$^{-1}$) | | | | | | |
| Aruana | 7250.3 ± 4290.6 | 6003.6 ± 3424.4 | 7913.7 ± 5719.1 | 7055.9 ± 4375.8 | | | |
| Marandu | 8362.6 ± 2031.3 | 7248.5 ± 865.7 | 7530.9 ± 885.8 | 7714.0 ± 1378.9 | | | |
| Mean | 7806.5 ± 3252.8 | 6626.1 ± 2468.5 | 7722.3 ± 3906.9 | 7385.0 ± 2877.4 | 0.638 | 0.563 | 0.808 |
| | Green leaf blade mass (kg DM/ha$^{-1}$) | | | | | | |
| Aruana | 1435.4 ± 621.9 | 1455.1 ± 630.8 | 1620.2 ± 445.1 | 1503.6 ± 544.3 | | | |
| Marandu | 2894.5 ± 768.3 | 2857.6 ± 642.0 | 3180.2 ± 620.7 | 2977.4 ± 655.9 | | | |
| Mean | 2165.0 ± 1012.3 | 2156.4 ± 951.1 | 2400.2 ± 963.8 | 2240.5 ± 600.1 | 0.563 | <0.01 | 0.952 |
| | Stalk mass (kg DM/ha$^{-1}$) | | | | | | |
| Aruana | 3813.1 ± 2779.7 | 3114.8 ± 2309.2 | 3827.1 ± 3319.0 | 3585.0 ± 2682.9 | | | |
| Marandu | 3200.8 ± 1257.9 | 2674.3 ± 722.2 | 2561.7 ± 732.8 | 2812.3 ± 926.7 | | | |
| Mean | 3507.0 ± 2081.7 | 2894.6 ± 1647.3 | 3194.4 ± 2385.0 | 3198.6 ± 1804.8 | 0.778 | 0.280 | 0.880 |
| | Dead material mass (kg DM/ha$^{-1}$) | | | | | | |
| Aruana | 2001.7 ± 1007.3 | 1433.6 ± 539.0 | 2466.3 ± 2088.4 | 1967.2 ± 1362.1 | | | |
| Marandu | 2267.2 ± 1262.7 | 1716.4 ± 587.9 | 1788.9 ± 404.5 | 1924.2 ± 825.8 | | | |
| Mean | 2134.5 ± 1097.8 | 1575.0 ± 557.6 | 2127.6 ± 1477.1 | 1945.7 ± 1094.0 | 0.396 | 0.910 | 0.504 |
| | Green leaf blade offer (kg of DM/100 kg of BW) | | | | | | |
| Aruana | 6.5 ± 2.7 | 6.3 ± 2.6 | 7.0 ± 1.8 | 6.6 ± 2.3 | | | |
| Marandu | 14.8 ± 4.5 | 13.3 ± 5.0 | 14.5 ± 5.4 | 14.2 ± 4.7 | | | |
| Mean | 10.7 ± 5.6 | 9.8 ± 5.3 | 10.7 ± 5.5 | 10.4 ± 3.5 | 0.947 | <0.01 | 0.897 |
| | Green leaf blades/stalk mass | | | | | | |
| Aruana | 0.5 ± 0.3 | 0.6 ± 0.3 | 0.5 ± 0.3 | 0.5 ± 0.3 | | | |
| Marandu | 1.1 ± 0.7 | 1.1 ± 0.7 | 1.3 ± 0.5 | 1.2 ± 0.5 | | | |
| Mean | 0.8 ± 0.6 | 0.9 ± 0.5 | 0.9 ± 0.6 | 0.9 ± 0.5 | 0.811 | <0.01 | 0.923 |
| | Green leaf blade/dead material mass | | | | | | |
| Aruana | 0.7 ± 0.2 | 1.0 ± 0.1 | 0.9 ± 0.3 | 0.9 ± 0.2 | | | |
| Marandu | 1.9 ± 1.5 | 1.8 ± 0.7 | 1.8 ± 0.5 | 1.8 ± 0.9 | | | |
| Mean | 1.3 ± 1.2 | 1.4 ± 0.6 | 1.3 ± 0.6 | 1.4 ± 0.6 | 0.740 | <0.01 | 0.844 |

C: effect due to the concentrate; P: effect due to pasture; C × P: effect due to interaction.

The DMM showed no difference ($p > 0.05$) between the analyzed pastures. The GLBM/SM as well as the GLBM/DMM were significantly influenced by the type of forage ($p < 0.05$).

The pasture type effect was significant ($p < 0.05$) for all variables (Table 3), while the concentrate supplementation level did not affect ($p > 0.05$) the variables evaluated. This relationship is probably a reflection of the higher production of the GLBM influenced by the pasture type, which affects its proportion, a pattern that was not followed for the stem mass and DMM (%).

**Table 3.** Percentage (%) of green leaf blade mass (GLBM, %), stalk mass + sheath (SM + S, %), and dead material mass (DMM, %) from Aruana and Marandu grass under different levels of concentrate supplementation.

| Pasture (P) | Concentrate Levels (C) | | | Mean | *p*-Value | | |
| | 0% | 1.5% | 3% | | C | P | C × P |
|---|---|---|---|---|---|---|---|
| | Green leaf blade mass (%) | | | | | | |
| Aruana | 22.6 ± 8.6 | 26.4 ± 6.8 | 25.1 ± 8.6 | 24.7 ± 7.7 | | | |
| Marandu | 36.8 ± 14.2 | 39.8 ± 9.2 | 42.6 ± 8.7 | 39.7 ± 10.6 | | | |
| Mean | 29.7 ± 13.4 | 33.1 ± 10.4 | 33.8 ± 12.3 | 32.2 ± 9.2 | 0.542 | <0.01 | 0.866 |
| | Stalk mass + sheath (%) | | | | | | |
| Aruana | 48.2 ± 11.7 | 47.4 ± 10.8 | 45.4 ± 8.2 | 47.0 ± 9.8 | | | |
| Marandu | 37.7 ± 9.0 | 36.9 ± 8.6 | 33.8 ± 7.3 | 36.1 ± 8.0 | | | |
| Mean | 42.9 ± 11.3 | 42.2 ± 10.8 | 39.6 ± 9.5 | 41.6 ± 8.9 | 0.669 | <0.01 | 0.985 |
| | Dead material mass (%) | | | | | | |
| Aruana | 29.1 ± 5.2 | 26.1 ± 5.1 | 29.3 ± 5.3 | 28.2 ± 5.1 | | | |
| Marandu | 25.3 ± 9.1 | 23.1 ± 5.6 | 23.5 ± 2.6 | 24.0 ± 6.0 | | | |
| Mean | 27.2 ± 7.3 | 24.6 ± 5.3 | 26.4 ± 5.0 | 26.1 ± 5.6 | 0.537 | 0.040 | 0.829 |

C: effect due to the concentrate; P: effect due to pasture; C × P: effect due to interaction.

### 3.2. Animal Performance and Diet Consumption

The DM intake in relation to body weight (DMI, % BW) was the only variable that did not present an interaction effect ($p > 0.05$) between the variables related to intake (Table 4). The total dry matter intake (TDMI, kg/day) and DMI (% BW) were not associated ($p > 0.05$) with the pasture type, while all variables were significantly influenced by the concentrate supplement level ($p < 0.05$).

**Table 4.** Feed and individual nutrients intake of lambs from Aruana and Marandu grass under different levels of concentrated supplementation.

| Pasture (P) | Concentrate Level (C) | | | Mean | *p*-Value | | |
| | 0% | 1.5% | 3% | | C | P | C × P |
|---|---|---|---|---|---|---|---|
| | Total Intake (DM g/day) | | | | | | |
| Aruana | 0.695 ± 0.10 [B] | 0.828 ± 0.07 [A] | 0.855 ± 0.05 [A] | 0.792 ± 0.10 | <0.01 | 0.082 | <0.01 |
| Marandu | 0.473 ± 0.01 [B] | 0.888 ± 0.07 [A] | 0.831 ± 0.07 [A] | 0.731 ± 0.17 | | | |
| Mean | 0.584 ± 0.14 [b] | 0.858 ± 0.07 [a] | 0.843 ± 0.06 [a] | 0.762 ± 0.13 | | | |
| | Total Intake (% BW) | | | | | | |
| Aruana | 2.41 ± 0.39 | 2.63 ± 0.32 | 2.64 ± 0.59 | 2.56 ± 0.44 | 0.020 | 0.713 | 0.218 |
| Marandu | 2.07 ± 0.07 | 3.04 ± 0.61 | 2.76 ± 0.78 | 2.62 ± 0.69 | | | |
| Mean | 2.24 ± 0.32 [b] | 2.83 ± 0.52 [a] | 2.70 ± 0.66 [a] | 2.59 ± 0.56 | | | |

**Table 4.** *Cont.*

| Pasture (P) | Concentrate Level (C) | | | Mean | *p*-Value | | |
|---|---|---|---|---|---|---|---|
| | 0% | 1.5% | 3% | | C | P | C × P |
| | Crude Protein Intake (g/day) | | | | | | |
| Aruana | 0.095 ± 0.01 [BC] | 0.140 ± 0.01 [A] | 0.158 ± 0.01 [A] | 0.131 ± 0.03 | <0.01 | <0.01 | <0.01 |
| Marandu | 0.040 ± 0.01 [C] | 0.111 ± 0.01 [B] | 0.150 ± 0.01 [A] | 0.100 ± 0.05 | | | |
| Mean | 0.067 ± 0.03 [c] | 0.125 ± 0.02 [b] | 0.154 ± 0.01 [a] | 0.116 ± 0.03 | | | |
| | Neuter Detergent Fiber Intake (g/day) | | | | | | |
| Aruana | 0.490 ± 0.06 [A] | 0.473 ± 0.04 [A] | 0.355 ± 0.03 [B] | 0.439 ± 0.08 | <0.01 | <0.01 | <0.01 |
| Marandu | 0.311 ± 0.01 [B] | 0.490 ± 0.04 [A] | 0.353 ± 0.03 [B] | 0.385 ± 0.08 | | | |
| Mean | 0.400 ± 0.10 [b] | 0.482 ± 0.04 [a] | 0.354 ± 0.03 [c] | 0.412 ± 0.08 | | | |
| | Acid Detergent Fiber Intake (g/day) | | | | | | |
| Aruana | 0.205 ± 0.02 [A] | 0.183 ± 0.02 [A] | 0.108 ± 0.01 [C] | 0.165 ± 0.05 | <0.01 | <0.01 | <0.01 |
| Marandu | 0.138 ± 0.01 [B] | 0.193 ± 0.02 [A] | 0.110 ± 0.01 [BC] | 0.147 ± 0.04 | | | |
| Mean | 0.171 ± 0.04 [b] | 0.188 ± 0.02 [a] | 0.109 ± 0.01 [c] | 0.156 ± 0.03 | | | |
| | Organic Matter Intake (g/day) | | | | | | |
| Aruana | 0.646 ± 0.09 [BC] | 0.741 ± 0.06 [AB] | 0.695 ± 0.05 [AB] | 0.694 ± 0.08 | <0.01 | 0.032 | <0.01 |
| Marandu | 0.438 ± 0.01 [C] | 0.790 ± 0.07 [A] | 0.715 ± 0.06 [AB] | 0.647 ± 0.16 | | | |
| Mean | 0.542 ± 0.13 [b] | 0.765 ± 0.07 [a] | 0.705 ± 0.05 [a] | 0.671 ± 0.12 | | | |

C: effect due to the concentrate; P: effect due to pasture; C × P: effect due to interaction. [a–c] Different letters on the same line indicate significant differences when the *p* value for C (concentrate). [A–C] Different letters on the same line indicate significant differences when the *p* value for C × P is *p* < 0.05.

The TDMI was lower in the animals that did not receive supplementation, regardless of the pasture. However, when comparing grasses, those that were kept on Marandu grass had significantly lower consumption (*p* < 0.05) than those kept on Aruana grass.

The CPI was lower (0.040 kg/day) for the animals grazing on Marandu grass without supplementation compared to the value for those grazing on Aruana without supplementation (0.090 kg/day), showing the deficiency of *Brachiaria* pasture in achieving the lambs' nutritional needs.

The lambs without supplementation did not receive enough intake to achieve their nutritional requirements for proper ADG, which was the most evident in the lambs grazing on Marandu grass.

The digestibility results did not show a significant interaction effect (*p* > 0.05) between the pasture and concentrate supplementation (Table 5).

**Table 5.** Apparent digestibility of dry matter (%), crude protein (%), neuter detergent fiber (%), acid detergent fiber (%), and organic matter (%) from Aruana and Marandu grasses under different levels of concentrate supplementation.

| Pasture (P) | Concentrate Level (C) | | | Mean | *p*-Value | | |
|---|---|---|---|---|---|---|---|
| | 0% | 1.5% | 3% | | C | P | C × P |
| | Dry matter (%) | | | | | | |
| Aruana | 46.7 ± 0.08 | 60.2 ± 0.03 | 54.3 ± 0.01 | 53.7 ± 0.07 | | | |
| Marandu | 54.6 ± 0.03 | 61.2 ± 0.03 | 57.0 ± 0.03 | 57.6 ± 0.04 | | | |
| Mean | 50.7 ± 0.07 [c] | 60.6 ± 0.03 [a] | 55.6 ± 0.02 [b] | 55.7 ± 0.05 | <0.01 | <0.01 | 0.120 |
| | Crude protein (%) | | | | | | |
| Aruana | 36.3 ± 0.10 | 60.5 ± 0.02 | 56.2 ± 0.02 | 52.7 ± 0.13 | | | |
| Marandu | 36.3 ± 0.09 | 57.3 ± 0.06 | 63.50 ± 0.05 | 51.4 ± 0.14 | | | |
| Mean | 35.8 ± 0.09 [b] | 58.9 ± 0.05 [a] | 59.8 ± 0.05 [a] | 52.0 ± 0.13 | <0.01 | 0.433 | 0.156 |

**Table 5.** *Cont.*

| Pasture (P) | Concentrate Level (C) | | | Mean | *p*-Value | | |
|---|---|---|---|---|---|---|---|
| | 0% | 1.5% | 3% | | C | P | C × P |
| Neuter detergent fiber (%) | | | | | | | |
| Aruana | 54.2 ± 0.06 | 52.3 ± 0.03 | 34.7 ± 0.05 | 47.0 ± 0.10 | | | |
| Marandu | 55.8 ± 0.05 | 55.0 ± 0.06 | 37.3 ± 0.04 | 49.4 ± 0.10 | | | |
| Mean | 55.0 ± 0.05 [a] | 53.7 ± 0.04 [a] | 36.0 ± 0.04 [b] | 48.2 ± 0.10 | <0.01 | 0.149 | 0.956 |
| Acid detergent fiber (%) | | | | | | | |
| Aruana | 50.5 ± 0.04 | 51.0 ± 0.03 | 35.8 ± 0.04 | 45.8 ± 0.08 | | | |
| Marandu | 51.5 ± 0.02 | 48.0 ± 0.01 | 29.2 ± 0.08 | 42.9 ± 0.11 | | | |
| Mean | 51.0 ± 0.03 [b] | 49.5 ± 0.03 [b] | 32.5 ± 0.07 [a] | 44.3 ± 0.09 | <0.01 | 0.055 | 0.114 |
| Organic matter (%) | | | | | | | |
| Aruana | 51.3 ± 0.07 | 61.5 ± 0.02 | 54.3 ± 0.01 | 55.7 ± 0.06 | | | |
| Marandu | 59.2 ± 0.03 | 62.5 ± 0.03 | 56.2 ± 0.03 | 59.3 ± 0.04 | | | |
| Mean | 55.2 ± 0.07 [b] | 62.0 ± 0.03 [a] | 55.2 ± 0.02 [a] | 57.5 ± 0.05 | <0.01 | 0.01 | 0.079 |

C: effect due to the concentrate; P: effect due to pasture; C × P: effect due to interaction; [a–c] different letters on the same line indicate significant differences for C.

The digestibility of total dry matter and organic matter was significantly influenced ($p < 0.05$) by the pasture type, while all variables were significantly influenced by the level of concentrate supplementation ($p < 0.05$) (Table 5).

In addition, there were differences in the NDF and ADF apparent digestibility in this study that were affected by supplementation ($p < 0.05$). The BCS was the only variable that showed significant interaction ($p < 0.05$) among the variables related to performance (Table 6). The ADG, TWG, and FC were significantly influenced ($p < 0.05$) by the pasture type, while all variables, except for the IBW, were significantly influenced by the concentrate supplementation level ($p < 0.05$).

**Table 6.** Initial body weight (kg), final body weight (kg), slaughter body weight (kg), empty body weight (kg), hot carcass weight (kg), average daily gain (kg/day), food conversion (kg DM/kg ADG), and body condition score (1–5) from Aruana and Marandu grasses under different levels of concentrate supplementation.

| Pasture (P) | Concentrate Level (C) | | | Mean | *p*-Value | | |
|---|---|---|---|---|---|---|---|
| | 0% | 1.5% | 3% | | C | P | C × P |
| Initial Body Weight (kg) | | | | | | | |
| Aruana | 22.64 ± 3.48 | 22.46 ± 3.37 | 22.46 ± 3.11 | 22.5 ± 3.12 | | | |
| Marandu | 23.03 ± 2.51 | 22.55 ± 3.13 | 22.50 ± 1.93 | 22.7 ± 2.40 | | | |
| Mean | 22.83 ± 3.03 | 22.50 ± 3.10 | 22.48 ± 2.47 | 22.6 ± 2.76 | 0.962 | 0.877 | 0.990 |
| Final Body Weight (kg) | | | | | | | |
| Aruana | 34.1 ± 3.31 | 39.3 ± 4.28 | 36.6 ± 4.45 | 36.7 ± 4.38 | | | |
| Marandu | 26.9 ± 6.73 | 38.6 ± 3.80 | 36.8 ± 4.67 | 34.1 ± 6.33 | | | |
| Mean | 30.5 ± 5.6 [b] | 38.9 ± 3.97 [a] | 36.7 ± 4.35 [a] | 35.4 ± 5.36 | <0.01 | 0.114 | 0.166 |
| Slaughter body Weight (kg) | | | | | | | |
| Aruana | 30.3 ± 3.08 | 36.2 ± 4.07 | 32.9 ± 4.36 | 33.2 ± 4.41 | | | |
| Marandu | 24.2 ± 6.77 | 35.1 ± 3.22 | 33.4 ± 4.55 | 30.9 ± 5.96 | | | |
| Mean | 27.2 ± 5.17 [b] | 35.6 ± 3.55 [a] | 33.2 ± 4.26 [a] | 32.0 ± 5.19 | <0.01 | 0.141 | 0.238 |

**Table 6.** *Cont.*

| Pasture (P) | Concentrate Level (C) | | | Mean | *p*-Value | | |
|---|---|---|---|---|---|---|---|
| | 0% | 1.5% | 3% | | C | P | C × P |
| Empty Body weight (kg) | | | | | | | |
| Aruana | 24.5 ± 2.68 | 31.2 ± 3.57 | 28.4 ± 3.30 | 28.0 ± 4.13 | | | |
| Marandu | 18.9 ± 5.68 | 28.8 ± 2.44 | 28.6 ± 3.86 | 25.4 ± 5.32 | | | |
| Mean | 21.7 ± 4.49 [b] | 29.9 ± 3.16 [a] | 28.5 ± 3.42 [a] | 26.7 ± 4.72 | <0.01 | 0.047 | 0.219 |
| Hot Carcass Weight (kg) | | | | | | | |
| Aruana | 13.1 ± 1.87 | 17.3 ± 2.33 | 15.3 ± 2.58 | 15.2 ± 2.80 | | | |
| Marandu | 9.8 ± 3.52 | 15.9 ± 1.90 | 15.6 ± 2.72 | 13.7 ± 3.44 | | | |
| Mean | 11.4 ± 1.83 [b] | 16.6 ± 2.16 [a] | 15.4 ± 2.53 [a] | 14.5 ± 3.12 | <0.01 | 0.101 | 0.287 |
| Average Daily Gain (kg/day) | | | | | | | |
| Aruana | 0.108 ± 0.01 | 0.183 ± 0.03 | 0.190 ± 0.06 | 0.160 ± 0.05 | | | |
| Marandu | 0.030 ± 0.04 | 0.163 ± 0.02 | 0.186 ± 0.05 | 0.126 ± 0.07 | | | |
| Mean | 0.069 ± 0.04 [b] | 0.173 ± 0.03 [a] | 0.188 ± 0.06 [a] | 0.143 ± 0.06 | <0.01 | 0.029 | 0.141 |
| Food Conversion (kg DM/kg ADG) | | | | | | | |
| Aruana | 12.0 ± 3.87 | 6.2 ± 0.96 | 7.1 ± 2.29 | 8.4 ± 3.62 | | | |
| Marandu | 16.3 ± 5.10 | 7.6 ± 1.28 | 7.4 ± 1.93 | 10.4 ± 5.23 | | | |
| Mean | 14.1 ± 4.8 [b] | 6.9 ± 1.29 [a] | 7.2 ± 2.03 [a] | 9.4 ± 4.42 | <0.01 | <0.01 | 0.432 |
| Body Condition Score (1–5) | | | | | | | |
| Aruana | 2.4 ± 0.20 [A] | 3.0 ± 0.44 [A] | 2.8 ± 0.40 [A] | 2.7 ± 0.42 | | | |
| Marandu | 1.5 ± 0.85 [B] | 2.8 ± 0.41 [A] | 3.3 ± 0.51 [A] | 2.4 ± 0.95 | | | |
| Mean | 1.9 ± 0.77 [b] | 2.9 ± 0.43 [a] | 3.0 ± 0.48 [a] | 2.6 ± 0.60 | <0.01 | 0.120 | 0.030 |

C: effect due to the concentrate; P: effect due to pasture; C × P: effect due to interaction. DM = dry matter. [a,b] Different letters on the same line indicate significant differences for C. [A,B] Different letters on the same line indicate significant differences when the *p* value for C × P is $p < 0.05$.

The animals' BCSs without supplementation and grazing on Marandu grass were significantly lower than those in the other treatments ($p < 0.05$).

### 3.3. Economic Analysis

The direct costs had a greater share of the total production cost. The total production costs for the supplementation levels of 0%, 1.5%, and 3% on Aruana and Marandu grasses were USD 881.13, 952.75, 935.10, 974.22, 1011.13, and 960.39, respectively (Table 7).

**Table 7.** Economic analysis for increasing levels of lamb supplementation in Aruana and Marandu grasses.

| Parameters | Aruana | | | Marandu | | |
|---|---|---|---|---|---|---|
| | 0% | 1.5% | 3% | 0% | 1.5% | 3% |
| Total lambs | 6 | 6 | 6 | 3 | 6 | 6 |
| Direct cost (USD) | 598.38 | 706.52 | 725.39 | 640.03 | 746.63 | 750.68 |
| Indirect cost (USD) | 282.74 | 246.23 | 209.71 | 334.17 | 264.50 | 209.71 |
| Total production cost (USD) | 881.13 | 952.75 | 935.10 | 974.22 | 1011.13 | 960.39 |
| Gross revenue (USD) | 1074.57 | 1426.09 | 1257.33 | 362.32 | 1311.10 | 1278.98 |
| Gross profit (USD) | 193.45 | 473.34 | 322.23 | −611.90 | 299.97 | 343.86 |
| Gross margin (%) | 34.72 | 64.03 | 49.44 | −325.80 | 44.14 | 51.87 |
| Days in the experiment | 202.55 | 175.55 | 148.54 | 243.06 | 189.05 | 148.54 |
| Carcass produced (kg) | 151.34 | 200.86 | 177.09 | 56.62 | 184.65 | 180.14 |
| Cost per kg of carcass (USD) | 11.23 | 9.14 | 10.19 | 33.20 | 10.57 | 10.01 |
| Cost per concentrate (USD) | - | 139.76 | 177.96 | - | 141.05 | 180.60 |
| Average daily gross profit (USD) | 1.85 | 5.21 | 4.19 | −4.86 | 3.07 | 4.13 |
| Accounting breakeven point (kg) | 95.22 | 71.43 | 72.84 | - | 90.80 | 74.64 |
| Accounting breakeven point (USD) | 676.20 | 507.25 | 517.30 | - | 644.81 | 529.96 |

Hot carcass price (BRL/kg) = 20.00; USD 1 = BRL 5.44 (4 April 2020).

## 4. Discussion

### 4.1. Quantitative Forage Estimate

The observed production of the total forage masses of 7055.92 and 7714.06 kg/ha for Aruana and Marandu grasses, respectively, was higher than that found by Carvalho et al. [27], who worked with Marandu grass in the Brazilian midwest region during the dry season, and higher than that reported by Emerenciano Neto et al. [28] regarding grazing with Aruana grass in the Brazilian northeast region.

The lower GLBM for Aruana grass (1503.63 kg DM/ha) compared to that for Marandu grass (2977.49 kg DM/ha) can probably be attributed to the lower relation to the GLBM/SM of Aruana grass compared to that of Marandu grass. In addition, Marandu grass has a greater leaf blade length than that of Aruana grass.

In addition, considering the lambs' grazing habits and greater selectivity, they probably grazed the tenderest parts of the Marandu grass leaves, providing greater leaf blade accumulation. Fajardo et al. [4], who worked with supplementation levels of 0%, 1.5%, and 2%, and Souza et al. [5], who utilized different leaf offers and only evaluated lambs and dairy sheep, observed a GLBM value for Aruana grass that was similar to that obtained in this study.

A lower GLBM (kg DM/ha) in Marandu grass was reported by Carvalho et al. [27] and Emerenciano Neto et al. [28] compared to that observed in this study. The undesirable elongation of SM (kg DM/ha) in the Aruana cultivar, which is justified by the smaller size of its leaves compared to those of Marandu grass, probably influenced the higher SM percentage in Aruana grass compared to that in Marandu grass.

The experiment timing coincided with high rainfall (874 mm), which probably influenced the lack of significant difference in the DMM between the pastures. The GLBS was three to four times above the animals' ingestion capacities, thus ensuring a balanced supply in quantity, as recommended by Hodgson [29]. The lowest supply of green leaf blades was recorded by Fajardo et al. [4], who studied lambs and dairy sheep in Aruana grass with four levels of supply of green leaf blades (4, 7, 10, and 13 kg DM/100 kg PV). In the dry season, Carvalho et al. [27] observed lower results for Marandu grass compared to those found in this study.

The values for GLBM/SM + Sheath were 0.58 and 1.22 for the Aruana and Marandu grasses, respectively, and those for the GLBM/DMM were 0.90 and 1.87, respectively. The Aruana grass showed a lower GLBM/SM than that of Marandu grass because the Aruana grass stalk is thinner and lighter than the Marandu grass stalk, which would indicate favorable conditions for leaf blade selection by lambs. Fajardo et al. [4] observed a similar relationship for the GLBM/SM in three evaluation periods with lambs receiving concentrate supplementation in Aruana grass. A longer grazing time leads to a lower GLBM/SM and tends to decrease the animal performance efficiency.

According to Brâncio et al. [30], the GLBM/SM is a very important tool for forage plant management. This is considered a critical limit when the values are less than 1.0, which implies a reduction in the quantity and quality of the produced forage, a situation verified in Aruana pasture grazing, probably due to the lower GLBM. The highest GLBM was registered for Marandu grass compared to that for Aruana grass, which could be explained by the larger plant size and climate adaptation.

The GLBM was influenced by the pasture type; this is probably a reflection of what affects its proportion, a pattern that was not followed for stem mass and DMM (%). Souza et al. [5] observed a higher percentage of GLBM and a similar SM to those obtained in this study in sheep grazing on Aruana grass. This is probably because the authors worked with a fixed GLBM offer. A higher percentage was found by Emerenciano Neto et al. [28] for the DMM post-grazing for Aruana and Marandu grasses, indicating greater senescent material loss left during grazing by the sheep.

It is important to remember that sheep systems maintained mainly by pastures are the main pillar of the economy in countries such as New Zealand, with around 55% of the total annual export earnings generated by the livestock industry. Pasture supply and pasture

fodder compose greater than 95% of the diets on New Zealand farms, and this system is efficient, sustainable, and relatively low-cost [31].

### 4.2. Animal Performance and Feed Consumption

The total dry matter intake (TDMI, kg/day) and DMI (% BW) were not influenced by the pasture type. The TDMI was lower in the animals that did not receive supplementation, regardless of the pasture. The energy protein density of the diet is possibly related to the low TDMI in the non-supplemented animals, which were primarily grazing on Marandu grass. Concentrate supplementation influenced the increase in TDMI regardless of the pasture type.

The dry matter intake in relation to body weight observed in the supplemented lambs was within the recommended range [32]. The low consumption in the non-supplemented animals probably occurred due to fluctuations in the nutritional quality and morphological changes in the pasture during the experiment, in addition to the greater difficulty of lambs adapting to grazing and the ability to search for the quantity and quality of pasture, considering that weaning occurred near the beginning of the experiment. Similar data to those found in this study for DMI (% BW) was reported by Barbosa et al. [33] in Ile de France, Suffolk, and Santa Ines lamb breeds kept on Aruana grass without supplementation.

A deficiency in CPI in the diet decreases consumption. The NDFI, ADFI, and OMI were also lower for the animals grazing on Marandu grass without supplementation. This was probably related to the reduced microbial activity in the rumen, decreasing food efficiency because of low protein and energy intakes. Even without supplementation, the lambs grazing on Aruana grass consumed greater amounts of fiber, possibly benefiting from the pasture structure.

Decreased DM and CP digestibility were observed in the diets of the non-supplemented animals, which may have been due to the low availability of CP and high NDF in the diets. Adequate forage and concentrate proportions, and the chemical composition of the diet, are prerequisites for high digestibility [34], which are requirements that were not met in the treatments without supplementation.

According to McDonald et al. [34], the primary chemical food constituent that determines the digestion rate is the ADF. However, the high NDF content (greater than 55%; Table 5) limited lamb consumption in this study.

NDF components in forage are not homogeneous [35], and their rumen digestibility can vary from <25% to >75% [32]. In this study, the NDF values were 55%, 54%, and 36% for 0%, 1.5%, and 3% of concentrate supplementation, respectively. Any increase in protein intake can lead to an increase in the apparent CP digestibility [34], which was observed in this study as 0.58 and 0.59 g/100 g with 1.5% and 3% supplementation levels, respectively. However, these levels were below those recommended by the NRC [32].

Pastures with low quality, lower consumption of CP, and higher levels of NDF in the diet may affect the microbial activity in the rumen [34,36], which may be due to the higher rate of passage and decreased digestibility. According to Owens and Goetsh [37], a decrease in the particle size in diets with high concentrate levels promotes an increase in the digesta passage kinetics and the digestion process through the gastrointestinal tract.

Higher consumption (Table 4) and digestibility (Table 5) in lambs that received supplementation were related to the higher SW, HCW, ADG, TWG, FC, and BCS values. The animals that received concentrate supplementation displayed increased energy and protein levels in the diet, intensifying the rumen fermentative activity and potentially increasing the non-degradable rumen protein due to the greater digesta passage kinetics, leading to an increase in the lambs' feed efficiency, even with a relatively low CP digestibility.

The concentrate level influenced the number of days required to reach SW. The animals that received a 3% supplement were slaughtered first, followed by those that received 1.5%, and finally, those that did not receive a supplement. A similar pattern was observed by Archimede et al. [38] and Papi et al. [39]. Thus, greater body growth is observed when either a high concentrate amount is consumed or the animal utilizes additional minerals,

nitrogen, and metabolizable energy more effectively when these are deficient in the pasture for the desired production levels [40].

Barbosa et al. [33] reported similar results for the ADG on Aruana grass without supplementation. Fajardo et al. [4] reported an ADG value similar to that obtained in this study on Aruana grass with 0%, 1.5%, and 2% supplementation levels by weight. The treatments affected the ADG, which led to an increase in the number of days needed to reach the required BCS for slaughter in lambs without concentrate supplementation. The low weight gain observed in the treatments without supplementation can be explained by the low level of DM intake, low CP digestibility, and fluctuation in the nutritional composition of the pasture associated with the relatively high nutritional requirements of animals, particularly considering that they were young animals and in full growth.

An improved FC value was observed in the animals that received 1.5% and 3% concentrate supplementation levels (6.90 and 7.25, respectively). Lambs grazing on Aruana grass showed improved FC values (8.44) compared to those grazing on Marandu grass (10.43). These results were similar to those obtained by Archimede et al. [38], who found that adding concentrate to the diet caused a decrease in the FC (7.0 for lambs without supplementation and 6.0, 5.7, and 5.7 for the inclusion of 150, 300, and 600 g of concentrate/day, respectively). High values (about 9–10) were also observed by Mahgoub et al. [41] in lambs from Oman, and by Papi et al. [39] in Chall sheep (7.35–9.53); these high values were probably due to the animal category studied and the type of fibrous diet with low CP.

The BCS results were due to the lower ADG, which drastically decreased adiposity in the carcass. Similar results were obtained by Díaz et al. [42] for lambs finished on a pasture with BCSs of 1.79 and 2.05, although these results were not significant ($p > 0.05$). For lambs finished in confinement, a low score was obtained because weaning had a negative effect on the lambs' growth during the first two weeks from the beginning of the experiment.

### 4.3. Economic Analysis

Stivari et al. [43] revealed that production cost estimates and economic viability studies are fundamental for livestock activities and for the adequate characterization of a production system. Regarding the economic analysis results, direct costs had a greater share of the total production cost. The higher FC of the supplemented animals in relation to the non-supplemented ones, and the time, in days, needed to reach the slaughter BW in the animals with 0% supplementation (126 days for Marandu) influenced the higher total production cost. This was also partially attributed to economic losses due to currency conversion and the devaluation of the Brazilian real (BRL), which was reflected in the hot carcass sale value of the animals without supplementation.

The higher ADG of the supplemented animals (1.5% and 3%) made it possible to reach slaughter in less time, decreasing the production cost in relation to that without supplementation. The higher hot carcass cost per kilogram of the non-supplemented animals (0%) on Marandu grass was partly due to the real (BRL) devaluation at the slaughter time for carcass sale, as well as due to the high FC, lower ADG, lower hot carcass production (kg), longer time to finish, and, mainly, animal death in this treatment. The deaths probably occurred due to the lower immunity of the animals from the low-nutrient diet provided by Marandu grass, demonstrating the need to use concentrate supplementation for lambs on pasture with this species of grass. In this sense, according to Vega-Britez et al. [44] and Melo et al. [45], lambs without supplementation finishing in Brachiaria grass present direct losses; with the death of the lambs, however, supplementation increases performance and reduces mortality, and at high levels, it is efficient in reducing the economic impact of intoxication by Brachiaria grass during finishing.

The lambs that grazed on Aruana grass with 1.5% supplementation showed the most positive economic result among the six systems analyzed. Higher net profit, gross margin, and amount of hot carcass produced were obtained in this system, in addition to lower cost per kg BW and the lowest accounting breakeven point. Moreover, this system presented the highest rates when compared with the other systems and had the highest values of gross

profit and average daily net (Table 7). Rozanski et al. [46] obtained greater economic gains when finishing lambs in feedlot with diet supplementations between 1.0 and 1.5% DM.

During an experiment with a Suffolk sheep herd, Stidivari et al. [43] observed that the expenditure on food (pasture, corn silage, and concentrated feed) was the largest contributor to the formation of variable costs within all systems at approximately 38%.

## 5. Conclusions

Exclusively finishing lambs on *Brachiaria brizantha* cv. Marandu grass without access to concentrate supplementation is not recommended, as animals take longer to reach the sale weight, representing increased production costs and reflecting negatively on animal performance. Finishing lambs on an Aruana pasture with a 1.5% BW concentrate supplementation showed improved productive and economic results.

**Author Contributions:** Conceptualization, G.D.V.-B., M.R. and F.M.d.V.J.; methodology, G.D.V.-B., A.M.d.S., A.L.A.d.S., C.M.C. and M.R.d.S.; validation, G.D.V.-B., M.R. and F.M.d.V.J.; formal analysis, G.D.V.-B., A.M.d.S., C.G.d.S. and F.M.d.V.J.; investigation, G.D.V.-B., M.R., A.L.A.d.S., C.M.C. and M.R.d.S.; resources, M.R. and F.M.d.V.J.; data curation, F.M.d.V.J.; writing—original draft preparation, G.D.V.-B., M.R. and F.M.d.V.J.; writing—review and editing, G.D.V.-B., C.G.d.S. and F.M.d.V.J.; visualization, M.R. and F.M.d.V.J.; supervision, M.R. and M.R.d.S.; project administration, F.M.d.V.J.; funding acquisition, F.M.d.V.J. All authors have read and agreed to the published version of the manuscript.

**Funding:** This research was funded by Fundação de Apoio ao Desenvolvimento do Ensino, Ciência e Tecnologia do Estado de Mato Grosso do Sul (FUNDECT); Coordenação de Aperfeiçoamento de Pessoal de Nível Superior (Capes—PROAP and DS Scholarship); Conselho Nacional de Desenvolvimento Científico e Tecnológico (CNPQ—PQ); Universidade Federal da Grande Dourados (UFGD—PAP).

**Institutional Review Board Statement:** Not applicable.

**Informed Consent Statement:** Not applicable.

**Data Availability Statement:** Data are contained within the article.

**Acknowledgments:** We would like to thank the support received by the Ovinotecnia research group (dgp.cnpq.br/dgp/espelhogrupo/8112331595527307) and those responsible for the laboratories/field area of the Federal University of Grande Dourados and EMBRAPA—CPAO, who were fundamental in carrying out this search.

**Conflicts of Interest:** The authors declare no conflicts of interest.

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
