# Peer review of "Effects of Using Different Concentrate Supplementation Levels in Diets of Lambs Fed Tropical Aruana (Megathyrsus maximus) or Marandu (Brachiaria brizantha) Grass: Performance, Digestibility, and Costs of Production"

_2813-3463, doi:10.3390/grasses3010003_

Round 1
Reviewer 1 Report
Comments and Suggestions for Authors
The study evaluated the effect of supplementation of lambs on two different forage types on the body weight gain, intake, digestibility and production cost. I have a few comments;
1-2; The title needs to be revised, authors should add what they are supplementing “supplementation of lambs”
--Authors need to include more information about the vegetative state of the grasses, including if the grasses were just planted/existing, indicate management of the grasses, manure addition, fertilizer, irrigatation etc.
162; Only one lamb per paddock was fitted with faecal collection bag, this is not ideal, does not give room for animal variation
All the tables in the results section are not well presented, difficult to follow
Author Response
Research paper | Manuscript ID: grasses-2709552
Supplementation of Lambs on Tropical Aruana (Megathyrsus maximus) or Marandu (Brachiaria brizantha) Grass: Body Weight Gain, Intake, Digestibility, and Production Costs
Review 1
Open Review
(x) I would not like to sign my review report
( ) I would like to sign my review report
Quality of English Language
(x) I am not qualified to assess the quality of English in this paper
( ) English very difficult to understand/incomprehensible
( ) Extensive editing of English language required
( ) Moderate editing of English language required
( ) Minor editing of English language required
( ) English language fine. No issues detected
Yes |
Can be improved |
Must be improved |
Not applicable |
|
Does the introduction provide sufficient background and include all relevant references? |
( ) |
( ) |
(x) |
( ) |
Are all the cited references relevant to the research? |
(x) |
( ) |
( ) |
( ) |
Is the research design appropriate? |
( ) |
(x) |
( ) |
( ) |
Are the methods adequately described? |
(x) |
( ) |
( ) |
( ) |
Are the results clearly presented? |
( ) |
( ) |
(x) |
( ) |
Are the conclusions supported by the results? |
( ) |
( ) |
( ) |
( ) |
Comments and Suggestions for Authors
RESPONSE:
Dear reviewer, the authors thank you for your important suggestions for improving the manuscript and below we have inserted the responses to each of the considerations and, when necessary, points have been inserted or adjusted in the manuscript with yellow markings.
Review 1 : The study evaluated the effect of supplementation of lambs on two different forage types on the body weight gain, intake, digestibility and production cost. I have a few comments;
1-2; The title needs to be revised, authors should add what they are supplementing “supplementation of lambs”
RESPONSE: We adjusted the title to: "Effect of using different concentrate level in diet of lambs, fed tropical Aruana (Megathyrsus maximus) or Marandu (Brachiaria brizantha) grass: performance, digestibility, and costs of production"
Review 1: Authors need to include more information about the vegetative state of the grasses, including if the grasses were just planted/existing, indicate management of the grasses, manure addition, fertilizer, irrigatation etc.
RESPONSE: The grasses had already been established for 3 years, were not fertilized or irrigated during the experimental phase and grazing was fixed. This information is in the material and methods.
Review 1: 162; Only one lamb per paddock was fitted with faecal collection bag, this is not ideal, does not give room for animal variation.
RESPONSE: The measurements carried out with the animals with collection bags were repeated measurements over time, as we observed in the results of the repetitions, this was effective in reducing the individual variation of the animal.
Review 1: All the tables in the results section are not well presented, difficult to follow
RESPONSE: We understand the difficulty, but the number of parameters presented is large, which ends up causing a saturation of visual information. In an attempt to minimize this, we reduced the excessive number of decimal places in the averages with the intention of reducing the amount of information.

Reviewer 2 Report
Comments and Suggestions for Authors
Supplementation of lambs on tropical Aruana (Megathyrsus maximus) or Marandu (Brachiaria brizantha) grass: Body weight gain, intake, digestibility, and production costs
Dear Authors,
The manuscript is interesting and describes effect different concentrate levels in lambs diet on the performance, the digestibility and the costs of productions in pasture systems with two different grass species. There is a lot of elements to correct, especially the statistical analysis, because in the subsection 2.9. Experimental Design and Statistical Analysis are described 3x2 design suggesting two-way ANOVA, but equation describes one-way ANOVA model. Tables must be corrected. Please check the technical words in the text in the available literature, ie. feed instead of food.
Additionally more present literature in introduction is needed.
Below I added some suggestions helpful during revision of manuscript:
Line 2
Lamb supplementation is not entirely accurate statement, it may be better to change from beginning to: ‘Effect of use different concentrate level in diet of lambs, fed tropical Aruana (Megathyrsus maximus) or Marandu (Brachiaria brizantha) grass: performance, digestibility, and costs of production’.
Line 21
Highlights are not needed in Grasses Journal, only the abstract is enough (although this is not a bad idea, as is the Simple Summary in some MDPI journals, i.e. Animals), all sentences from Highlights are included in Abstract.
Line 51
More references are needed in the Introduction (one at least for each sentence).
Line 110
Table 1, title. ‘The proportion of concentrate feed ingredients and chemical compositions of grazing simulation’ maybe better to modified it as: The proportion of ingredients in concentrate and chemical composition components of diet (%/kg of feed).
Line 111
In text is dry matter %, maybe better to use % in kg of feed, because dry matter of concentrate is 87% (+13% of water).
Line 150
Subsection title 2.4 Food Analysis must be changed for 2.4. Feed Analysis
The same situation in line 160: 2.5 Feed Consumption and Digestibility.
Please check in whole manuscript context of use this word: food – in case of humans, feed – in case of animals…. line 168, line 171…
Line 192
In the text of the manuscript is described that slaughtering took place at 73,77,91,98,105 126th day of experiment. How the final weight was calculated and statistics? Because when in experiment lambs was allocated randomly to 12 paddocks (36 non-castrated males), that gives 3 lambs in the paddock. When slaughtering procedure was conducted at 6 mentioned dates, there will left no animals after third slaughtering, that needs explanation. The number of animals will be too low to obtain appropriate power of a test. In future needed will be to increase number of lambs or decrease number of slaughtering dates or both. In this experiment are:
One way design – 6 treatments (0%A, 0%M, 1.5%A, 1.5%M, 3%A, 3%M) each with 2 replications = 12 paddocks x 3 lambs = 36 animals in total
Two way design – 3 treatments (concentrate level) each with 4 replications = 12 paddocks x 3 lambs
x
2 treatments (grass type) each with 6 replications = 12 paddocks x 3 lambs
Line 226
In subsection 2.9. Experimental Design and Statistical Analysis on line 227 it was described that the design with two 3x2 factors was used in experiment, also provided information about the use of analysis of variance. However, it has not been specified whether it was one-way or two-way ANOVA. Equation of variance model refers to one-way ANOVA. Information about the post-hoc test is also needed. The description of the methodology may also include information about presenting the results in the form of mean and standard deviation.
Line 235
In manuscript lack is table described Pearson correlation between 3 variables, mentioned in subsection 2.9.
Line 248-293
The tables’ titles must be abbreviated, without the standard deviation and the mean in the title's sentence, describing all examined parameters.
Line 250
Table 2 describes the yield obtained from pastures depending on the grass species. It cannot be combined with concentrate. There one factor can be used: kind of grass on pasture or second factor could be raw/green and wilted/dry matter (respectively for “total forage mass” and “dead material mass” used in text). Kg can be changed on tons.
p-value (sample) instead of P-value must be used (line 250-line 295)
In text below table: is Ns: not significant, p>0.1, must be p>0.05
Additionally dots must be used instead of comma in case of decimals.
Line 259
Table 3. In this case, a better solution will be one-way ANOVA, without interactions.
Line 267
Title of table 4 must be changed, ie.: Feed and individual nutrients intake.
Item |
Total intake (DM g/day) |
Total intake (% BW) |
CP intake (g/day) |
… |
Concentrate level (%) |
|
|
|
|
0 |
|
|
|
|
1.5 |
|
|
|
|
3.0 |
|
|
|
|
p-value |
|
|
|
|
Pasture |
|
|
|
|
Aruana |
|
|
|
|
Marandu |
|
|
|
|
p-value |
|
|
|
|
Interaction C x P |
|
|
|
|
a,b different superscripts in the same line (row) indicate significant differences: at p< 0.05; A,B at p<0.01
Or one -way ANOVA can be also used taking into a consider 6 different treatments: 0%A, 0%M, 1.5%A, 1.5%M, 3%A, 3%M
Line 284
Title of the table 5 must be changed
Information about significance added: a,b different superscripts in the same line (row) indicate significant differences at p< 0.05; A,B at p<0.01
Line 301
Table 7, row: total lambs.
36 heads in total, in this case to 0% Marandu was allocated 6 lambs.
Line 358
In text of manuscript is: ‘4.2 Animal Performance and Diet Consumption’ must be 4.2 Animal Performance and Feed Consumption.
Line 435 and 465
Paragraph starts 2 times with ‘Stivari et al. [39]’, in line 465 can be changed for ie.: ‘During experiment with Suffolk sheep herd Stidivari et al. [39]…’
Line 477
References
Abbreviation of Journal name:
· No. 18: Crop Sci.
· No. 19: J. Agric. Sci.
· No. 21: Semin. Ciênc. Agrár.
· No. 23: Semin. Ciênc. Agrár.
· No. 24: Biosci. J.
· No. 25: Grass Forage Sci.
· No. 41: Trop. Anim. Health Prod.
Author Response
Research paper | Manuscript ID: grasses-2709552
Supplementation of Lambs on Tropical Aruana (Megathyrsus maximus) or Marandu (Brachiaria brizantha) Grass: Body Weight Gain, Intake, Digestibility, and Production Costs
Review 2
Open Review
(x) I would not like to sign my review report
( ) I would like to sign my review report
Quality of English Language
(x) I am not qualified to assess the quality of English in this paper
( ) English very difficult to understand/incomprehensible
( ) Extensive editing of English language required
( ) Moderate editing of English language required
( ) Minor editing of English language required
( ) English language fine. No issues detected
Yes |
Can be improved |
Must be improved |
Not applicable |
|
Does the introduction provide sufficient background and include all relevant references? |
( ) |
( ) |
(x) |
( ) |
Are all the cited references relevant to the research? |
(x) |
( ) |
( ) |
( ) |
Is the research design appropriate? |
( ) |
( ) |
(x) |
( ) |
Are the methods adequately described? |
( ) |
( ) |
(x) |
( ) |
Are the results clearly presented? |
( ) |
( ) |
(x) |
( ) |
Are the conclusions supported by the results? |
( ) |
(x) |
( ) |
( ) |
Comments and Suggestions for Authors
RESPONSE: Dear reviewer, the authors thank you for your important suggestions for improving the manuscript and below we have inserted the responses to each of the considerations and, when necessary, points have been inserted or adjusted in the manuscript with yellow markings.
Review 2: The manuscript is interesting and describes effect different concentrate levels in lambs diet on the performance, the digestibility and the costs of productions in pasture systems with two different grass species. There is a lot of elements to correct, especially the statistical analysis, because in the subsection 2.9. Experimental Design and Statistical Analysis are described 3x2 design suggesting two-way ANOVA, but equation describes one-way ANOVA model. Tables must be corrected. Please check the technical words in the text in the available literature, ie. feed instead of food.
RESPONSE: Your suggestion was added to the text (manuscript), correcting elements in the statistical part and technical words.
Review 2: Additionally more present literature in introduction is needed.
RESPONSE: Add two more current bibliographic references to the manuscript.
Review 2: Below I added some suggestions helpful during revision of manuscript:
Line 2
Lamb supplementation is not entirely accurate statement, it may be better to change from beginning to: ‘Effect of use different concentrate level in diet of lambs, fed tropical Aruana (Megathyrsus maximus) or Marandu (Brachiaria brizantha) grass: performance, digestibility, and costs of production’.
RESPONSE: we agreed with the suggestion of the new title and inserted it into the manuscript.
Review 2: Line 21
Highlights are not needed in Grasses Journal, only the abstract is enough (although this is not a bad idea, as is the Simple Summary in some MDPI journals, i.e. Animals), all sentences from Highlights are included in Abstract.
RESPONSE: we deleted Highlights.
Review 2: Line 51
More references are needed in the Introduction (one at least for each sentence).
RESPONSE: we added two more current bibliographic references in the introduction of the manuscript.
Review 2: Line 110
Table 1, title. ‘The proportion of concentrate feed ingredients and chemical compositions of grazing simulation maybe better to modify it as: The proportion of ingredients in concentrate and chemical composition components of diet (%/kg of feed).
RESPONSE: suggestion accepted and adjusted in the manuscript.
Review 2: Line 111
In text is dry matter %, maybe better to use % in kg of feed, because dry matter of concentrate is 87% (+13% of water).
RESPONSE: suggestion accepted and adjusted in the manuscript.
Review 2: Line 150
Subsection title 2.4 Food Analysis must be changed for 2.4. Feed Analysis
RESPONSE: suggestion accepted and adjusted in the manuscript.
Review 2: The same situation in line 160: 2.5 Feed Consumption and Digestibility.
Please check in entire manuscript context of use this word: food – in case of humans, feed – in case of animals…. line 168, line 171…
RESPONSE: suggestion accepted and adjusted in the manuscript, feed to feed.
Review 2: Line 192
In the text of the manuscript is described that slaughtering took place at 73,77,91,98,105 126th day of experiment. How the final weight was calculated and statistics? Because when in experiment lambs was allocated randomly to 12 paddocks (36 non-castrated males), that gives 3 lambs in the paddock. When slaughtering procedure was conducted at 6 mentioned dates, there will left no animals after third slaughtering, that needs explanation. The number of animals will be too low to obtain appropriate power of a test. In future needed will be to increase number of lambs or decrease number of slaughtering dates or both. In this experiment are:
One way design – 6 treatments (0%A, 0%M, 1.5%A, 1.5%M, 3%A, 3%M) each with 2 replications = 12 paddocks x 3 lambs = 36 animals in total
Two way design – 3 treatments (concentrate level) each with 4 replications = 12 paddocks x 3 lambs x 2 treatments (grass type) each with 6 replications = 12 paddocks x 3 lambs
RESPONSE: We inserted the following more explanatory text into the manuscript: " The lambs were slaughtered based on body condition score (BCS) 2.5–3.0 (scale 1–5) (Kenyon et al., 2014), or at 6 months of age, following a finishing/physiological pattern of tissue deposition (Osório et al., 2012) regardless of treatment, resulting in slaughters at 73, 77, 91, 98, 105 and 126 experimental days, with an average of six animals per day. The animals were slaughtered 110 km from the experimental site, in an experimental slaughterhouse at the Federal University of Grande Dourados."
Review 2: Line 226
In subsection 2.9. Experimental Design and Statistical Analysis on line 227 it was described that the design with two 3x2 factors was used in experiment, also provided information about the use of analysis of variance. However, it has not been specified whether it was one-way or two-way ANOVA. Equation of variance model refers to one-way ANOVA. Information about the post-hoc test is also needed. The description of the methodology may also include information about presenting the results in the form of mean and standard deviation.
RESPONSE: The text in the manuscript was readjusted by inserting the suggestions.
Review 2: Line 235
In manuscript lack is table described Pearson correlation between 3 variables, mentioned in subsection 2.9.
Line 248-293
The tables’ titles must be abbreviated, without the standard deviation and the mean in the title's sentence, describing all examined parameters.
RESPONSE: The suggestion was inserted in the text of the manuscript.
Review 2: Line 250
Table 2 describes the yield obtained from pastures depending on the grass species. It cannot be combined with concentrate. There one factor can be used: kind of grass on pasture or second factor could be raw/green and wilted/dry matter (respectively for “total forage mass” and “dead material mass” used in text). Kg can be changed on tons.
p-value (sample) instead of P-value must be used (line 250-line 295)
In text below table: is Ns: not significant, p>0.1, must be p>0.05
Additionally dots must be used instead of comma in case of decimals.
RESPONSE: The suggestions were accepted and inserted into the text of the manuscript.
Review 2: Line 259
Table 3. In this case, a better solution will be one-way ANOVA, without interactions.
RESPONSE: A one-way Anova was performed and there was no effect, so we chose to leave it as originally proposed.
Review 2: Line 267
Title of table 4 must be changed, ie.: Feed and individual nutrients intake.
Item |
Total intake (DM g/day) |
Total intake (% BW) |
CP intake (g/day) |
… |
Concentrate level (%) |
|
|
|
|
0 |
|
|
|
|
1.5 |
|
|
|
|
3.0 |
|
|
|
|
p-value |
|
|
|
|
Pasture |
|
|
|
|
Aruana |
|
|
|
|
Marandu |
|
|
|
|
p-value |
|
|
|
|
Interaction C x P |
|
|
|
|
a,b different superscripts in the same line (row) indicate significant differences: at p< 0.05; A,B at p<0.01
Or one -way ANOVA can be also used taking into a consider 6 different treatments: 0%A, 0%M, 1.5%A, 1.5%M, 3%A, 3%M
Line 284
Title of the table 5 must be changed
Information about significance added: a,b different superscripts in the same line (row) indicate significant differences at p< 0.05; A,B at p<0.01
Line 301
Table 7, row: total lambs.
36 heads in total, in this case to 0% Marandu was allocated 6 lambs.
RESPONSE: All adjustments requested in the text, titles and footers of the tables were made directly in the manuscript.
Review 2: Line 358
In text of manuscript is: ‘4.2 Animal Performance and Diet Consumption’ must be 4.2 Animal Performance and Feed Consumption.
Line 435 and 465
Paragraph starts 2 times with ‘Stivari et al. [39]’, in line 465 can be changed for ie.: ‘During experiment with Suffolk sheep herd Stidivari et al. [39]…’
Line 477
References
Abbreviation of Journal name:
- No. 18: Crop Sci.
- No. 19: J. Agric. Sci.
- No. 21: Semin. Ciênc. Agrár.
- No. 23: Semin. Ciênc. Agrár.
- No. 24: Biosci. J.
- No. 25: Grass Forage Sci.
- No. 41: Trop. Anim. Health Prod.
RESPONSE: All adjustments requested in the text regarding standardization or correction were made directly in the manuscript and are identified.

Round 2
Reviewer 2 Report
Comments and Suggestions for Authors
Dear Authors
Most of the suggestions were incorporated in the revised manuscript. I only have two more suggestions:
Line 231: The equation is missing one more letter in subscript k.
In case of two-way ANOVA should be:
Yijk = μ + Pi + Cj + (PxC)ij + εijk
Line 270: It is difficult to determined using letters differences between each factor (effect of pasture and effect of concentrate level, and p-values for each factor). To present it transposition of table is needed, in this case will be also possible to add of interaction between each factor with p-value.
This kind of table is present in article: https://www.mdpi.com/2076-2615/9/5/253
Item |
Total intake (DM, g/day) |
… |
Pasture |
|
|
Aruana |
|
|
Marandu |
|
|
p-value |
|
|
Concentrate level |
|
|
0 |
|
|
1.5 |
|
|
3 |
|
|
p-value |
|
|
PxC, p-value |
|
|
Or is also possible to analyse data using one-way ANOVA
Item |
Total intake (DM, g/day) |
… |
Treatment |
|
|
|
|
|
Aruana, 0% concentrate |
|
|
Aruana, 1.5% concentrate |
|
|
Aruana, 3.0% concentrate |
|
|
Marandu, 0% concentrate |
|
|
Marandu, 1.5% concentrate |
|
|
Marandu, 3.0% concentrate |
|
|
p-value |
|
|
Author Response
Reviewer 2,
We appreciate suggestions/corrections.
Adjustments made directly to the manuscript: adjusted equation and tables.
We forward the corrected manuscript file with all adjustments made marked in yellow.
Yours sincerely,
